# Revealing brain cell-stratified causality through dissecting causal variants according to their cell-type-specific effects on gene expression

Ruo-Han Hao [1], Tian-Pei Zhang[1], Feng Jiang [1], Jun-Hui Liu [1], Shan-Shan Dong [1], Meng Li[2], Yan Guo [1] ✉ & Tie-Lin Yang [1,2] ✉

The human brain has been implicated in the pathogenesis of several complex diseases. Taking advantage of single-cell techniques, genome-wide association studies (GWAS) have taken it a step further and revealed brain cell-type-specific functions for disease loci. However, genetic causal associations inferred by Mendelian randomization (MR) studies usually include all instrumental variables from GWAS, which hampers the understanding of cell-specific causality. Here, we developed an analytical framework, Cell-Stratified MR (csMR), to investigate cell-stratified causality through colocalizing GWAS signals with single-cell eQTL from different brain cells. By applying to obesity-related traits, our results demonstrate the cell-type-specific effects of GWAS variants on gene expression, and indicate the benefits of csMR to identify cell-type-specific causal effect that is often hidden from bulk analyses. We also found csMR valuable to reveal distinct causal pathways between different obesity indicators. These findings suggest the value of our approach to prioritize target cells for extending genetic causation studies.

The human brain is one of the most complex tissues that have diversified functions. As the command center of nervous system, the brain takes parts in several biological processes. In addition to receiving and processing sensory information, maintaining normal behavior and cognition, the brain has recently attracted attention for its function in regulating complex traits. For example, the brain's contributions to maintain energy homeostasis and control metabolic traits have been widely recognized[1,2]. The abnormal regulation in brain therefore is regarded as one of the triggers for some diseases. Genome-wide association studies (GWAS) also supported the brain-related genetic etiology of complex traits. With numerous diseases/traits associated variants were identified, the next-step exploration of GWAS has been made to find tissues of relevance[3–5]. Notably, the brain was found significantly

enriched for not only psychological disorders but also other complex traits such as BMI, height, inflammatory bowel disease, etc[4–6], suggesting the general role of brain in maintaining body health.

Like other tissues, the human brain also exhibits cell type complexity. Current innovations in single-cell techniques, such as single-cell RNA sequencing (scRNA-seq) and single-cell assay for transposase accessibility by sequencing (scATAC-seq), have enabled profiling molecular features in individual cells, and revealed regulatory architectures in a cell type specific manner. Recently, Bryois et al. generated population-scale single-cell RNA-seq data and identified cell-type-specific eQTLs by associating genetic variants with gene expression of brain cells[7]. Integration with GWAS data further suggested cell-specific pathogenic effects of risk variants. This study provides us an

[1]Biomedical Informatics & Genomics Center, Key Laboratory of Biomedical Information Engineering of Ministry of Education, School of Life Science and Technology, Xi'an Jiaotong University, Xi'an, Shaanxi 710049, P. R. China. [2]Department of Orthopedics, The First Affiliated Hospital of Xi'an Jiaotong University, Xi'an, Shaanxi 710061, P. R. China. ✉e-mail: guoyan253@xjtu.edu.cn; yangtielin@xjtu.edu.cn

unprecedented opportunity to study the cell type choice of genetic variants to regulate gene expression.

Using genetic variants as instrumental variables, Mendelian randomization (MR) infers causal effects of exposure trait on outcomes in the presence of unobserved confounding. It is quite useful to distinguish causality from observed associations without animal studies or randomized controlled trials[8]. Because of its time- and cost-efficient advantages, MR has become popular for assessing potentially causal relationships between risk factors and brain-related traits/diseases[9,10]. However, most of these MR studies use all genetic instrumental variables to make assumptions, considering the cell-type-specific nature of risk variants regulation, it is not clear which brain cell type contribute to the expected causality.

In this study, with the help of single-cell eQTL data, we sought to partition disease associated variants into different brain cells. Taking BMI as the trait of interest, we first investigated which cell type the causal variants affected gene expression in by profiling their cell-specific colocalization statuses. We next included cell-stratified SNPs in MR analysis to assess causal effects of BMI on other disease outcomes in terms of particular cell. We also developed a computational tool called Cell-Stratified Mendelian Randomization (csMR) to help automatically find cell-stratified causality by providing GWAS summary data.

## Results

### The framework of cell-stratified MR analysis

An overview of csMR is shown in Fig. 1. Basically, csMR employs three steps to infer causality in each cell type: (1) finding genetic colocalization between GWAS and single-cell eQTL data after identifying causal variants with fine-mapping; (2) filtering the colocalized signals in each cell type to select suitable instrument variants (IVs); (3) performing two-sample MR analysis with selected IVs. Brain single-cell eQTL data were derived from Bryois et al[7]., which includes summary statistics for 8 cell types originated from brain cortex (excitatory neuron (ExN), inhibitory neuron (InN), astrocyte, microglia, oligodendrocyte (ODC), oligodendrocyte precursor cell (OPC), endothelial cell (EC) and pericyte). We adopted the SuSiE[11] method for fine-mapping and the "coloc" framework[12] for colocalization analysis under the assumption of multiple causal variants. Posterior probability of hypothesis 4 (PPH4), which indicates whether there is a shared causal association between GWAS and eQTL, was used to assess colocalization (Methods). In each cell type, we identified significant colocalization if PPH4 is greater than 0.8. The colocalized causal SNPs were then passed to the next step to select suitable IVs for MR analysis, which includes SNP filtering, clumping and quality control (Fig. 1). Two-sample MR analysis was finally conducted with cell-stratified IVs to assess the causal effects of exposure traits on outcomes in particular cell type context. We implemented six MR methods here to compensate for the intrinsic bias of each method, and prioritized the main MR method according to the results of pleiotropy analyses (Methods). Although this framework was initially designed to find cell-stratified causality using single-cell eQTL data, it is also compatible with other summary-level QTL datasets under different contexts. For example, we applied csMR with eQTL data from bulk tissues in the subsequent analyses. csMR is now publicly available at https://github.com/rhhao/csMR.

### The colocalization profiles of BMI suggesting cell-type-specific regulation of risk variants

We took BMI as the trait of interest to show how csMR partitioned genetic variants to different brain cells from the colocalization analysis. We obtained BMI GWAS summary data performed in 681,275 participants of European ancestry reported by Yengo et al., which was a meta-analysis from the Genetic Investigation of Anthropometric Traits (GIANT) consortium and the UK Biobank (Supplementary Data 1). 941 near-independent SNPs were found associated with BMI in the original

study (at a revised genome-wide significance threshold of $P < 1 \times 10^{-8}$). After colocalization analysis, as shown in Fig. 2a, we identified ~235 colocalized SNPs in brain cells with an average number of 171 (Supplementary Data 2). These SNPs were found to affect the expression of ~168 genes (120 genes on average; Supplementary Data 2). The proportion of cell-specific results ranged from 61.3% to 82.2% for colocalized SNPs and 74.2% to 91.3% for colocalized genes (Fig. 2a). We next investigated the average eQTL effects of colocalized SNPs in other cell types, and found that these SNPs had less obvious effect sizes in other test cells (Fig. 2b), suggesting that BMI-associated SNPs tend to regulate gene expression in a cell-type-specific manner. In addition, for colocalized genes, we also observed that 90.2% of these genes were recognized in a specific cell type (Fig. 2c). Next, we were interested to know how colocalized gene expressed in different brain cell types. We obtained scRNA-seq data of healthy human brain from Morabito et al[13]., which has generated gene expression matrix for astrocyte, ExN, InN, microglia, ODC and OPC. The pairwise comparison across cells suggested a cell-type-specific expression tendency for all colocalized genes and for those identified in a specific cell type (Fig. 2d; Supplementary Fig. 1 and Supplementary Data 3). Together, these results indicate that BMI-associated genetic variants would potentially act as brain eQTLs by affecting the expression of a specific gene in a specific cell type. Apart from BMI reported here, the cell-specific colocalization has also been observed for several brain disorders[7].

Here, we show an instance where the BMI-associated locus colocalized with eQTL derived from a specific brain cell type. As the locuszoom plots presented in Fig. 2e, within 4 cell types expressing *POMC*, we detected strong colocalization evidence only in InN (PPH4 = 1). The causal variant rs564667 was significantly associated with both BMI variation ($P = 6.99 \times 10^{-40}$) and *POMC* expression in InN ($P = 0.018$). *POMC* gene encodes Proopiomelanocortin that has long been studied for its role in energy homeostasis and body weight control[14,15]. We observed colocalization signals for *POMC* only in inhibitory neuron, which is consistent with its known neural regulation function[16].

### The cell-stratified causal effects of BMI inferred by Mendelian randomization

Given the cell-type-specific effects of risk variants indicated by colocalization analysis, csMR then used colocalized SNPs recognized in each cell as IVs and assessed cell-stratified causality of interested traits/diseases. For BMI, we explored the causal effects on 18 disorders belonging to 4 different disease categories that have previously shown relevance to BMI (Supplementary Data 4). We first performed MR analyses to infer the causal relationships between BMI and 18 outcomes using all associated variants with $P < 5 \times 10^{-8}$. The results showed that BMI was causally associated with almost all disease outcomes except for Alzheimer's disease and Parkinson's disease (Supplementary Data 5). Previous MR studies also concluded no causal association between BMI and Alzheimer's disease or Parkinson's disease[17], which was consistent with our results. We then focused on investigating whether the inferred causality was stratified by brain cells. IV selection was executed to filter colocalized SNPs. The SNPs strongly associated with BMI ($P < 5 \times 10^{-8}$) but not confounders (smoking, drinking, education) were further subjected to linkage disequilibrium (LD) pruning and outlier removal (Methods). The number of IVs ($N_{IV}$) that passed all quality control tests ranged from 4 to 34. After MR and additional pleiotropy and sensitivity analyses, we found significant causal effects of BMI on 7 complex diseases at $P < 2.78 \times 10^{-4}$ (0.05/180) (Fig. 3; Supplementary Fig. 2 and Supplementary Data 6 and 7), including sleep disorders (nervous system disease), attention deficit hyperactivity disorder (ADHD) (mental disorder), gout and osteoporosis (musculoskeletal disease), coronary artery disease (CAD), myocardial infarction and type II diabetes (T2D) (cardiovascular and metabolic disease). These causal relationships

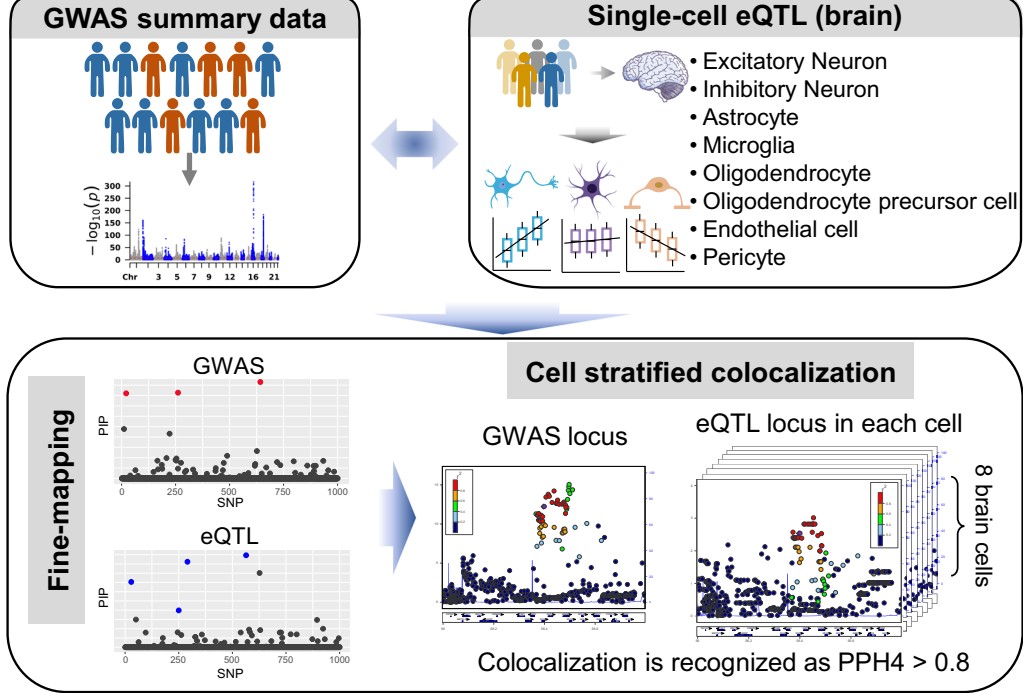

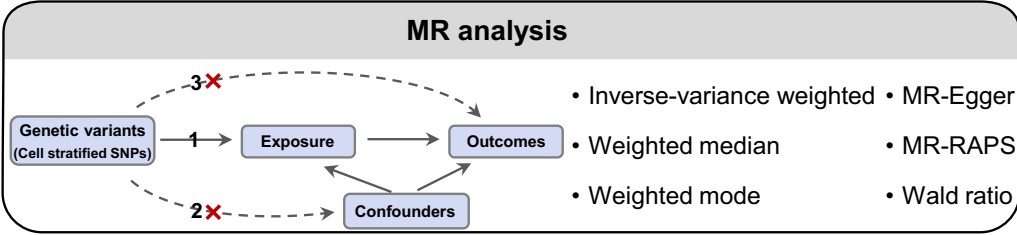

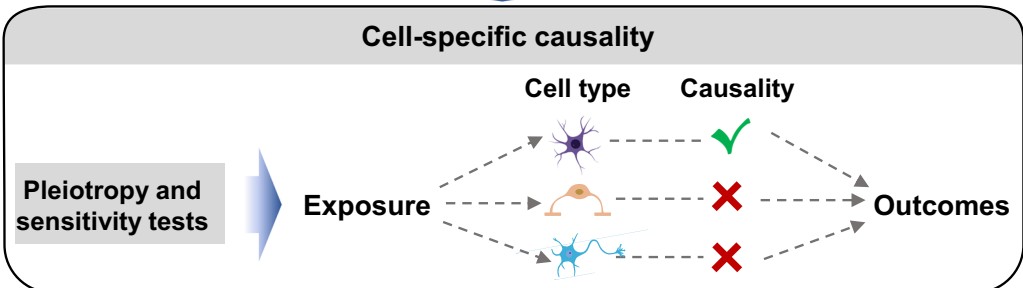

**Fig. 1 | Overview of csMR. Briefly, csMR took GWAS summary statistics and eQTL data as inputs.** Colocalization analysis was then implemented to identify colocalized variants with PPH4 > 0.8 in each cell type. Next, a stringent SNP filtering procedure was executed to select qualified instrumental variables in each cell type. MR analysis was conducted with cell-stratified variants using different methods. Finally, after correcting MR results with pleiotropy and sensitivity analyses, the causal relationship between exposure and outcome with respect to each cell type was drawn. Some figure elements were obtained from Servier Medical Art by Servier, which is licensed under a Creative Commons Attribution 4.0 License. Changes were made to the pictures.

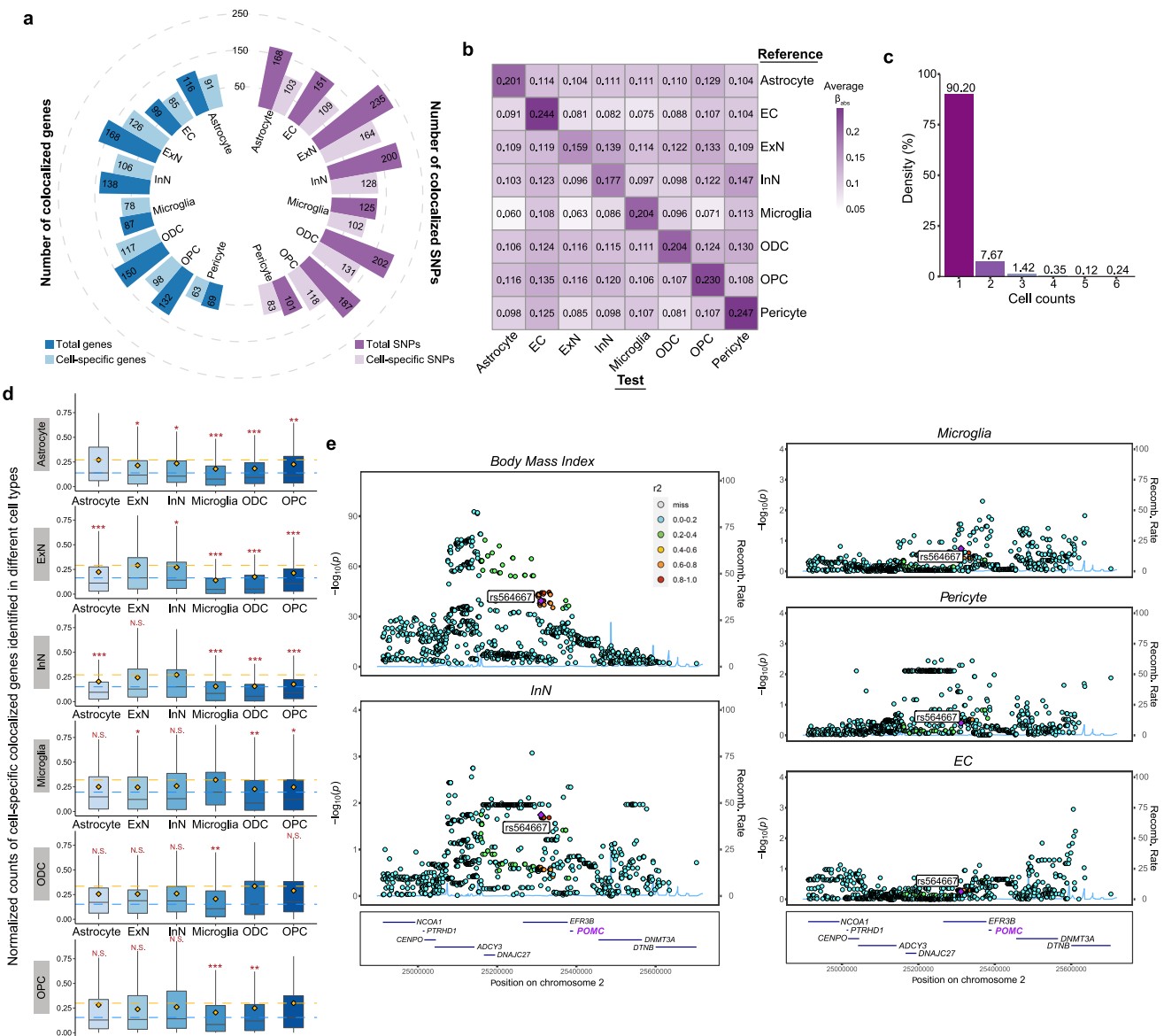

**Fig. 2 | Characterization of genetic colocalization between BMI-associated variants and gene expression in each brain cell type. a** The number of total and cell-specific colocalized SNPs and genes identified in each cell type. **b** Estimates of the average eQTL effect sizes in "test" cell types for colocalized SNPs identified in the "reference" cell types. **c** The proportion of colocalized SNPs identified in different number of cells. Most SNPs colocalized in a single cell type. **d** Comparison of normalized expression of cell-specific colocalized genes across different cell types. The "reference" cell types where genes colocalized are shown on the left. Each box plot shows the distribution of averaged expression (calculated from normalized read counts derived from a public scRNA-seq dataset[13]) of cell-specific colocalized genes in the "test" cell type. The box plots represent 25th, 50th, and 75th percentiles, and whiskers extend to 1.5 times the interquartile range. The yellow diamond indicates average expression in each test cell. The yellow and blue dashed lines show the mean and median expression levels in reference cell, respectively. The numbers of tested genes (*N*) are 82, 113, 99, 68, 104, 87, which were identified in reference cell astrocyte, ExN, InN, microglia, ODC and OPC, respectively. The gene expression in reference cell was compared with that in each test cell, and *P* values were calculated by two-sided paired Wilcoxon tests (*** *P* < 0.001, ** *P* < 0.01, * *P* < 0.05, "N.S." stands for not significant). The exact *P* values are listed in Data 3. **e** LocusZoom plots illustrating the association for variants at the *POMC* loci with BMI and *POMC* gene expression in different cells with detectable eQTL signals, including inhibitory neuron, microglia, pericyte and endothelial cell. The cell-specific causal variant is marked with purple diamond and annotated. The strength of their association with each trait is indicated by -log₁₀*P*.

were mainly observed in 6 brain cell types which are astrocyte, EC, ExN, microglia, ODC and OPC.

Notably, we found causal associations of BMI with sleep disorders, ADHD, gout, CAD and T2D in multiple cell types. In particular, higher BMI predicted by cell-stratified variants in EC, ExN, microglia, ODC and OPC was associated with increased risk of sleep disorders (inverse-variance weighting (IVW) beta [95% CI] per 1-SD increase in BMI: EC, 0.66 [0.37-0.94], $P = 7.61 \times 10^{-6}$, $N_{IV} = 19$; ExN, 0.84 [0.62-1.06], $P = 1.64 \times 10^{-13}$, $N_{IV} = 26$; microglia, 1.03 [0.61-1.46], $P = 2.23 \times 10^{-6}$, $N_{IV} = 14$; ODC, 0.58 [0.32-0.83], $P = 8.47 \times 10^{-6}$, $N_{IV} = 21$; OPC, 0.70 [0.44-0.96], $P = 1.16$ × $10^{-7}$, $N_{IV} = 24$). Positive association between BMI and ADHD was mainly observed in astrocyte and EC (IVW beta [95% CI]: astrocyte, 0.66 [0.36-0.97], $P = 1.85 \times 10^{-5}$, $N_{IV} = 22$; EC, 0.66 [0.34-0.98], $P = 5.70 \times 10^{-5}$, $N_{IV} = 21$). Additionally, we identified significant causal relationship between BMI and gout in ExN, microglia and ODC (ExN, 1.09 [0.65-1.52], $P = 8.32 \times 10^{-7}$, $N_{IV} = 34$; microglia, 1.71 [0.86-2.56], $P = 8.18 \times 10^{-5}$, $N_{IV} = 13$; ODC, 1.33 [0.77-1.89], $P = 3.19 \times 10^{-6}$, $N_{IV} = 21$). Moreover, cell-stratified positive causal association with CAD was observed in astrocyte and ExN (IVW beta [95% CI]: astrocyte, 0.21 [0.12-0.30], $P = 8.36 \times 10^{-6}$, $N_{IV} = 26$; ExN, 0.24 [0.15-0.33], $P = 1.56 \times 10^{-7}$, $N_{IV} = 29$).

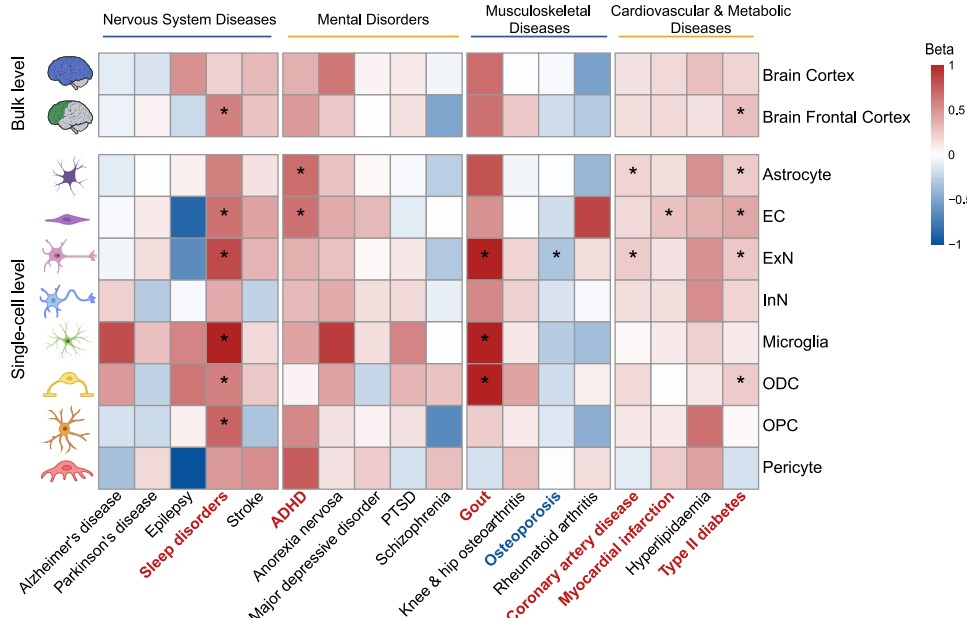

**Fig. 3 | Identifying causal relationships between BMI and 18 disease traits.**
Heatmap showing the cell-stratified and tissue-stratified causal effects of BMI on disease outcomes that were categorized into 4 disease classes. Causal associations were indicated by IVW MR analysis (beta coefficients and *P* values). Cells are colored according to the beta coefficients with red and blue corresponding to positive and negative associations, respectively. Bonferroni correction was used to adjust for multiple testing, and significant associations after adjustment ($P < 2.78 \times 10^{-4}$) are marked with asterisks. Disease traits that were positively and negatively associated with BMI are colored in red and blue, respectively. Some figure elements were obtained from Servier Medical Art by Servier and Scidraw.io (10.5281/zenodo.5348394), which are both licensed under a Creative Commons Attribution 4.0 License. Changes were made to the pictures.

We also found that higher genetically predicted BMI in astrocyte, EC, ExN and ODC were causally associated with higher risk of T2D (IVW beta [95% CI]: astrocyte, 0.24 [0.12-0.35], $P = 4.76 \times 10^{-5}$, $N_{IV} = 20$; EC, 0.41 [0.25-0.56], $P = 2.14 \times 10^{-7}$, $N_{IV} = 16$; ExN, 0.26 [0.16-0.35], $P = 1.59 \times 10^{-7}$, $N_{IV} = 27$; ODC, 0.25 [0.14-0.36], $P = 7.74 \times 10^{-6}$, $N_{IV} = 21$).These results emphasize that the influence of BMI on some complex diseases is potentially driven by the regulation consequences from multiple brain cells.

Besides, we detected cell-specific causality for osteoporosis and myocardial infarction. The higher risk of myocardial infarction was causally influenced by higher BMI estimated from EC stratified IVs (IVW beta [95% CI]: 0.27 [0.15-0.40], $P = 2.55 \times 10^{-5}$, $N_{IV} = 16$). While estimated from ExN stratified IVs, an increase of 1 SD in BMI was associated with lower risk of osteoporosis (IVW beta [95% CI]: −0.33 [−0.49 - −0.17], $P = 5.64 \times 10^{-5}$, $N_{IV} = 30$). These observations emphasize the cell-specific influence of BMI on osteoporosis and myocardial infarction.

### Stratifying causal relationships by single-cell eQTL finds more associations than using tissue-based data
We were interested to know whether the stratified causality identified using single-cell eQTL differed from that using tissue-based eQTL data. Since csMR is also compatible with other summary-level QTL results, we next applied csMR to GTEx eQTL data from bulk brain cortex and frontal cortex. As shown in Fig. 3, we successfully replicated causal effects of BMI on sleep disorders and T2D using stratified risk variants in brain frontal cortex (IVW beta [95% CI]: sleep disorders, 0.59 [0.34-0.83], $P = 2.79 \times 10^{-6}$, $N_{IV} = 26$; T2D, 0.30 [0.20-0.39], $P = 3.19 \times 10^{-9}$, $N_{IV} = 33$; Supplementary Fig. 3 and Supplementary Data 8 and 9). However, the associations with the other 5 diseases that had been pointed out using single-cell eQTL data were failed to reach a significant level after Bonferroni correction ($P < 2.78 \times 10^{-4}$), but only showed nominal significance ($P < 0.05$) for ADHD, gout, CAD and myocardial infarction. Besides, we didn't detect novel associations using tissue-based eQTL. These results suggest the advantage of integrating single-cell data to find hidden instructive information.

### Applying csMR to other obesity-related traits highlights causal effects on different outcomes
We next attempted to investigate how csMR performed on other traits. Although BMI is commonly used to approximate overall excess of body fat, there are other surrogates that also have been used to capture abnormal metabolic consequences related to adiposity. For example, the assessment of central adiposity can be indicated by waist-to-hip ratio (WHR), while measurement of body fat percentage can provide accurate fat mass percentage of adipose tissue. As the well accepted indicators of obesity, both BMI and other obesity-related traits were found related to brain[4,18,19], while the strongest genetic factors appeared to be largely distinct[20,21]. Here, we applied csMR to two additional obesity-related traits, WHR adjusted for BMI (WHRadjBMI) and body fat percentage, to see if they exhibited different cell-stratified causality to BMI. The colocalization results and suitable IV information are summarized in Supplementary Data 2, 11 and 13.

Among 18 outcomes, we finally recognized causal effects of WHRadjBMI on stroke, CAD and T2D (Fig. 4a, Supplementary Fig. 4 and Supplementary Data 10 and 11). The influence on stroke was not found from BMI. 3 effective cell types were identified to support the causal relationship between WHRadjBMI and stroke (IVW beta [95% CI]: astrocyte, 0.76 [0.42-1.09], $P = 9.71 \times 10^{-6}$, $N_{IV} = 16$; ExN, 0.66 [0.33-0.98], $P = 7.96 \times 10^{-5}$, $N_{IV} = 21$; InN, 0.61 [0.30-0.92], $P = 1.14 \times 10^{-4}$, $N_{IV} = 18$). Notably, although CAD and T2D were found associated with both BMI and WHRadjBMI, the effective cell types were distinct. The causal effect on CAD was observed in InN for WHRadjBMI (IVW beta [95% CI]: 0.30 [0.19-0.42], $P = 3.84 \times 10^{-7}$, $N_{IV} = 12$), while effects of BMI on CAD were observed in astrocyte and ExN. The influence on T2D were observed in InN and OPC for WHRadjBMI (IVW beta [95% CI]: InN, 0.24 [0.12-0.35], $P = 7.95 \times 10^{-5}$, $N_{IV} = 16$; OPC, 0.26 [0.13-0.39], $P = 7.61 \times 10^{-5}$, $N_{IV} = 12$), however, the associations between BMI and T2D were found in astrocyte, EC, ExN and ODC.

In addition, we recognized causal associations between body fat percentage and epilepsy, ADHD, schizophrenia and knee & hip

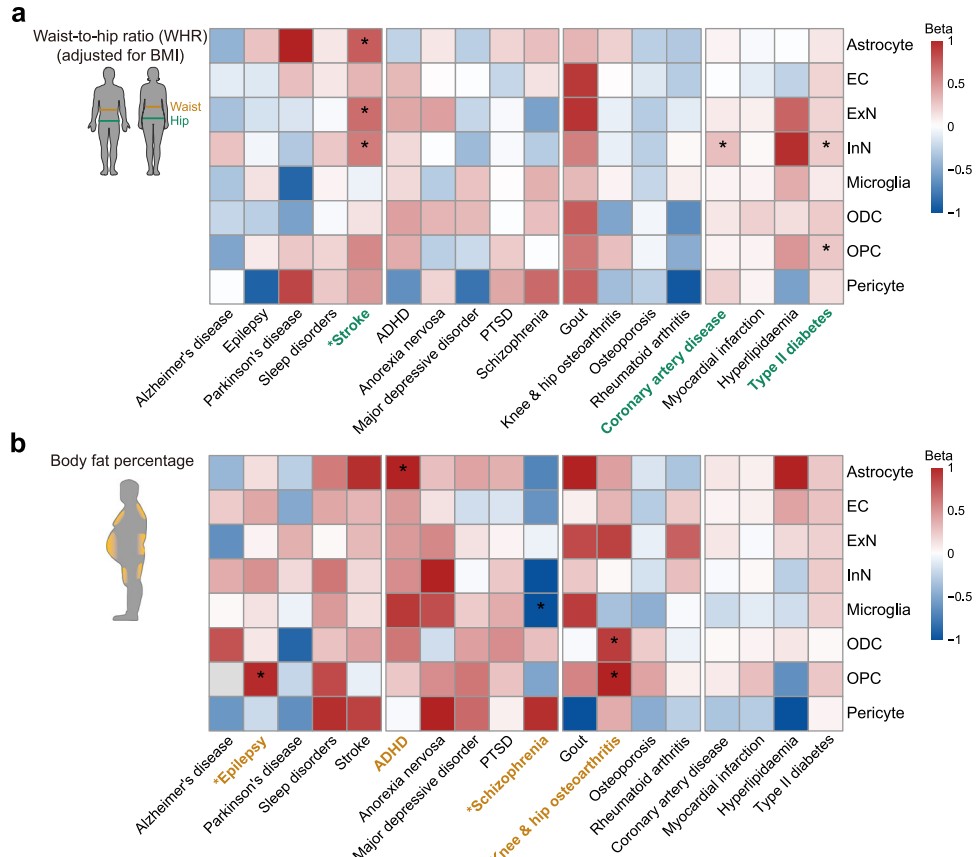

**Fig. 4 | Identifying causal relationships between WHRadjBMI, body fat percentage and 18 disease traits.** Heatmaps showing the cell-stratified effects of (**a**) WHRadjBMI and (**b**) body fat percentage on disease outcomes. Cells are colored according to the beta coefficients from IVW MR analysis and are marked with asterisks for significant associations after adjusting for multiple testing with Bonferroni correction ($P < 2.78 \times 10^{-4}$). Disease traits that were found significantly affected by WHRadjBMI or body fat percentage are colored in green and yellow, respectively, and those not affected by BMI are annotated with asterisks. Some figure elements in panel **a** were obtained from Servier Medical Art by Servier, which is licensed under a Creative Commons Attribution 4.0 License. Changes were made to the pictures.

osteoarthritis (Fig. 4b, Supplementary Fig. 5 and Supplementary Data 12 and 13). These influenced outcomes were not observed in the analyses focusing on BMI and WHRadjBMI, except for ADHD. Particularly, we found body fat percentage was positively associated with epilepsy in OPC (IVW beta [95% CI]: 0.97 [0.46-1.48], $P = 1.75 \times 10^{-4}$, $N_{IV} = 7$), and with knee & hip osteoarthritis in ODC and OPC (IVW beta [95% CI]: ODC, 0.89 [0.42-1.35], $P = 1.80 \times 10^{-4}$, $N_{IV} = 17$; OPC, 1.39 [0.68-2.10], $P = 1.18 \times 10^{-4}$, $N_{IV} = 8$), while negatively associated with schizophrenia in microglia (IVW beta [95% CI]: −1.08 [−1.58 - −0.57], $P = 3.14 \times 10^{-5}$, $N_{IV} = 9$). Together, these results suggest that different obesity measurements may genetically contribute to the incidence of different disease outcomes via brain-cell-specific regulation pathways.

## Discussion

Previous studies have revealed that disease risk variants identified by GWAS usually impart their phenotypic effect through regulating target genes in specific tissues[22]. However, little is known about the particular cell types where this functional regulation might take place. Besides, the causal effects inferred by all associated variants ignore the cell-specific regulation information. In this study, taking advantages of brain single-cell eQTL data, we developed a computational framework called csMR to help partition genetic effects into different cell types and assess the causal associations with disease outcomes using the cell-stratified variants. By applying csMR to GWAS summary data of BMI, our findings indicate cell-type-specific preference of BMI-associated variants by the evidence of colocalization, which highlights the

necessity to locate target cell types when exploring the functional effects for causal variants. Additionally, by performing two-sample MR analyses involving the partitioned variants as genetic instruments, we have identified cell-stratified causal associations. By comparing the stratified causal effects after colocalizing with tissue-level eQTL data, our results indicate that the associations identified through integrating single-cell eQTL data not only replicate the results drawn by tissue-level data, but can also recognize novel causal relationships. These observations suggest that selecting instrumental variants based on their cell-type-dependent effects helps elucidate the underlying causal influence on phenotypic variation that is usually hidden from bulk data. Meanwhile, dissecting associations into different cells helps better understand the functional cell types for inferred causalities. Finally, we have tested csMR on different obesity-related traits, and demonstrated its applicability in processing other GWAS summary data and drawing credible results.

The results of two-sample MR analyses highlight the underlying function of glial cell types (astrocytes, microglia, ODC and OPC) in contributing to the pleiotropic effects of BMI associated variants especially on sleep disorders, ADHD, gout, CAD and T2D. The causal associations of obesity with these disease conditions have been elucidated by previous MR studies[23–27], while the effective cell types are still poorly understood. The glial cells are essential for maintaining the brain tissue homeostasis and neuronal integrity. Recently, emerging evidence has revealed their profound role on energy homeostasis regulation, especially on obesity pathogenesis[28–31]. It is also demonstrated that the abnormal metabolic milieu, in turn, can disrupt the

brain-to-periphery communication mediated by glial cells, which leads to disease conditions such as glucose intolerance and increased blood pressure[2]. In this study, we pinpointed glial cells as the responsive cells to BMI variation in which corresponding gene expression alterations increase the risk of sleep disorders, ADHD, gout, CAD and T2D. The role of glial cells in regulation of sleep has been widely revealed, which includes regulating sleep-wake cycles and influencing sleep pressure, duration as well as intensity[32,33]. It has also been proved that astrocyte drives ADHD through neuron-astrocyte communication[34]. As for T2D, significant roles of astrocyte[35] and oligodendrocyte[36] in central control of diabetic phenotypes have also attracted attention, which is in line with our findings. We also found significant effects of BMI on gout and CAD after instrumented with eQTLs in glia cells. Though direct association between astrocyte and CAD has not been fully elucidated, the underlying mechanisms could be explained by the predominant function of astrocyte in controlling blood glucose and blood pressure[35], of which the increase are regarded as major risk factors to develop CAD. As for gout, evidence from recent observational studies suggest the association between gout and brain structure[37], however, the mechanism underlying how brain or which brain cell type affects gout is still unclear.

It is also worth noting that we observed causal effects of BMI on 5 disease outcomes (sleep disorder, gout, osteoporosis, CAD and T2D) restricted to excitatory neuron. As we mentioned above, causal associations between BMI and sleep disorder, gout, CAD, T2D were previously demonstrated. Neuronal regulation has been proven to be fundamental in the control of obesity[38,39], sleep disorder[40], T2D[41] and CAD[42], supporting the reliability of our results. In addition, we also prioritized excitatory neuron as the target cell for the associations with osteoporosis, which was observed as 1 SD higher genetically predicted BMI was associated with a decreased osteoporosis risk. While the association between BMI and osteoporosis has been elucidated by previous MR analysis[43], here, we found that the effect is particularly exhibited in excitatory neuron. Studies have shown sufficient evidence for the control of bone metabolism[44] via neuronal regulation. Intriguingly, there is a close link between bone and fat metabolism that is modulated by neuroendocrine. A bunch of neuroregulators were recognized to coordinately regulate bone mass and adiposity, such as the neuropeptide Y (NPY)[45] and leptin[46]. Though the complex mechanisms connecting obesity and osteoporosis have not been fully understood, our results offer a potential genetic explanation, which the effects are most likely mediated by gene expression variation in excitatory neurons.

Additionally, we found genetic variants associated with BMI and gene expression in endothelial cell have independent effect on increased risk for sleep disorder, ADHD, myocardial infarction and T2D, which implies the participation of endothelial cells in mediating between obesity and these complications. Gene dysregulation in brain endothelial cell was found associated with obesity by leveraging single-cell RNA-seq data in mice model[47]. As the key constituent of the blood-brain barrier, the function of endothelial cell in regulating sleep disorder, ADHD, myocardial infarction and T2D mainly involves restricting access of circulatory factors that might be the pathogenesis of these complications[48,49]. Although previous findings support the significant role of endothelial cells in developing obesity and other diseases, these studies, however, were conducted independently. Question remains whether obesity-related alteration in endothelial cells causes comorbidities. Here, we conclude endothelial-cell-specific causal relationships following MR paradigm after executing a strict quality control procedure, which can ensure the reliability and robustness of our results. Whereas, the underlying biological mechanisms behind these findings need to be further elucidated.

By comparing the MR results of different obesity indicators, we observed causal effects on either distinct disease outcomes or same outcomes in different cell types, suggesting different causal pathways between various surrogate obesity indicators. The inconsistent results could be explained by the fact that these traits reflect different aspects of body composition, which was proven to be affected by distinct molecular processes and metabolic mechanisms. BMI, estimated from height and weight, mainly reflects overall adiposity, while WHR, measured by waist and hip circumferences, considers both abdominal fat and gluteofemoral fat. Unlike BMI that does not distinguish fat mass from lean mass, body fat percentage directly measures the amount of fat and provides more accurate assessment of adiposity. Previous GWAS studies showed that the majority of associated loci and affected genes were unique between these traits[50,51]. They also depicted different metabolic pathways, which BMI-associated loci were significantly enriched for appetite regulation, whereas pathways involved in alterations in fat cell size and body fat amount were identified by body fat percentage associated loci[52]. In addition, studies have revealed distinct proteomic profiles associated with BMI and WHR[53,54], indicating distinguishing molecular processes. Together, the differences of biological processes, potentially activated in specific cell, could be the possible interpretation of the inconsistent cell-stratified causal effects between different obesity indicators.

The original brain single-cell eQTL data only included cis-eQTL. cis-eQTLs usually reflect direct effects on genes, whereas trans-eQTLs expose sets of downstream genes and pathways on which the effects of disease variants converge. Further exploration with available cell-specific trans-eQTL data would help illustrate the causal effects of genetic variants acting in trans. It is worth mentioning that, though we used cis-eQTL to select IVs, csMR also accepts summary-level trans-eQTL with no need to make any modifications.

Overall, we developed an analytical framework, csMR, to find cell-stratified causality by integrating summary-level data of GWAS and single-cell eQTL. After applying it to BMI GWAS data, we showed the application value of csMR that help prioritize target cells for inferred causal associations between BMI and 18 disease outcomes, which is often hidden from bulk analyses. Our method is time- and cost-efficient and will contribute to better understanding of the cell-specific genetic link between complex diseases.

## Methods
### Overview of csMR
csMR was developed based on the Snakemake workflow management system[55]. Figure 1 presents an overview of the analytical framework. csMR is now publicly available at https://github.com/rhhao/csMR and Supplementary Software 1.

### Datasets
**eQTL data.** In this study, we dissected genetic variables into different brain cell types by integrating single-cell eQTL data. Summary-level brain single-cell eQTL data were obtained from a study conducted by Bryois et al[7]. This is currently the only eQTL study of all major cell types in the adult human brain by performing single-cell RNA sequencing. In brief, the single-cell eQTL dataset contains eQTL results in 8 cell types (astrocytes, endothelial cells, excitatory neurons, inhibitory neurons, microglia, oligodendrocytes, oligodendrocyte precursor cells and pericytes) mainly from brain prefrontal and temporal cortices of 192 European individuals. The expression of ~14,595 genes were quantified, and 5.3 million SNPs were genotyped in total according to the descriptions in the original study.

In order to see whether dissecting genetic variables using single-cell eQTLs outperforms using bulk eQTLs on finding more underlying causal relationships, we additionally obtained eQTL results derived from bulk brain tissues reported by GTEx (v8)[56]. We used bulk eQTLs from brain cortex and frontal cortex in this study as they were generated from similar brain regions as the single-cell eQTLs. The sample sizes are also comparable with single-cell eQTL study. In each tissue, about 10.4 million SNPs were tested for association with the expression

of over 20,000 genes. Detailed information of eQTL datasets used in this study can be found in Supplementary Data 1.

**GWAS data of obesity-related traits.** Here, we tested the performance of csMR on GWAS datasets of obesity-related traits. Firstly, we obtained GWAS summary statistics of BMI from a meta-analysis of previous GWAS from the Genetic Investigation of Anthropometric Traits (GIANT) consortium and the UK Biobank[57] with a total sample size of 681,275 individuals of European ancestry. We also included summary-level GWAS data of two additional obesity indicators, WHRadjBMI[58] and body fat percentage (https://gwas.mrcieu.ac.uk/datasets/ukb-b-8909/), which contain 694,649 and 454,633 European individuals, respectively. According to the descriptions of sample inclusion in these original studies, over 60% of subjects were from UK Biobank. Therefore, we used genome data of 50,000 European individuals from UK Biobank as the reference panel in subsequent analyses. We constructed this reference data by sampling unrelated European ancestry individuals based on the genetic kinship information (Data-Field 22021), ethnic background information (Data-Field 21000) and genetic ethnic grouping information (Data-Field 22006) available from the UK Biobank (https://biobank.ndph.ox.ac.uk/). Further information of obesity GWAS data resources is summarized in Supplementary Data 1.

**GWAS data of outcomes.** To investigate whether there were cell-stratified causal effects of obesity on complex diseases, we performed Mendelian randomization analyses taking 18 disease traits as outcomes that were previously proven to have close relationship with obesity from either observational or Mendelian randomization studies. These outcomes were further classified into 4 categories including nervous system diseases, mental disorders, musculoskeletal diseases, cardiovascular and metabolic diseases, according to the disease class terms defined by Medical Subject Headings (MeSH) (https://meshb-prev.nlm.nih.gov/). Each category contains 4-5 disease traits and we have summarized related publications in Supplementary Data 4 supporting their relationships with obesity. GWAS summary statistics of disease traits were obtained from different resources as listed in Supplementary Data 4. All the studies claimed that there were no overlapping individuals from the UK Biobank and GIANT consortiums, and all the participating samples or > 80% of the samples are European. The sample sizes for these studies ranged from 9954 to 446,696.

**Colocalization analyses.** Colocalization approaches have been successful in addressing the problem of finding shared causal variants between a molecular trait (e.g. gene expression) and a disease trait. In this study, we stratified genetic effects by applying colocalization strategy to GWAS and single-cell eQTL data to identify causal variants that were associated with both GWAS trait and gene expression in a particular cell type. We performed colocalization analyses using the updated "coloc" package (version 5)[12], which adopted the SuSiE approach[11] for fine mapping before colocalization. The SuSiE method is recommended by the authors of coloc as it outperformed other methods especially in the situation when multiple causal variants exist[12]. LD matrix was generated based on a reference panel comprising 50,000 unrelated individuals in the UK Biobank as mentioned above. Variants within the MHC region were excluded (chr6:28477797-33448354, GRCh37 assembly). Credible sets were identified by setting "coverage = 0.9". We used default parameters to evaluate colocalization between GWAS signals and the expression of genes within 100 kb flanking regions. The posterior probabilities were finally estimated to support 5 competing hypotheses: (1) no association with either trait (PPH0); (2) association with trait 1 only (PPH1); (3) association with trait 2 only (PPH2); (4) association with both traits, but have different causal variants (PPH3); (5) association with both traits with common causal variants (PPH4). Evidence for colocalization between GWAS and gene expression was then assessed using PPH4. A generally accepted threshold (PPH4 > 0.8) was used to find strong evidence of colocalization[59,60]. The outputs of "coloc" also include putative causal variants for both GWAS and eQTL associations. The GWAS hits with strong evidence of colocalizing with single-cell eQTL were further regarded as candidate instrumental variables for subsequent MR analysis.

**Genetic instrumental variables selection.** We next selected qualified SNPs from the causal variants adopted from colocalization analysis for MR analysis. A qualified genetic instrumental variable should satisfy the three core assumptions[61]: (1) it should be strongly associated with the exposure (relevance assumption); (2) it should not be associated with any potential confounders of the exposure-outcome association (independence assumption); (3) it is associated with the outcome only through the exposure (exclusion restriction assumption). According to these assumptions, we first selected SNPs that were strongly associated with exposure ($P < 5 \times 10^{-8}$). Variants were further filtered out by checking if they or their proxy variants ($r^2 > 0.8$) were associated with confounders using "phenoscanner" v1.0 R package[62]. Three potential confounders were considered, including education, drinking and smoking behavior. We next identified index SNPs from the remaining instruments using LD clumping procedure provided by PLINK[63] through setting $r^2$ threshold of 0.001, a window size of 10,000 kb and a $P$ value threshold of $5 \times 10^{-8}$. These SNPs were then intersected with outcome data. For those not presented in the outcome dataset, we replaced them by their proxies ($r^2 > 0.8$) that were included. Next, data harmonization was performed using "TwoSampleMR" v0.5.6 R package[64] to ensure that the effects on the exposure and outcome corresponded to the same alleles. Meanwhile, palindromic SNPs with intermediate allele frequencies (> 0.42) were removed.

**Quality control of instrumental variables.** For the instruments satisfying the three MR assumptions, we execute the quality control processes to assure the instrument quality. We first implemented heterogeneity test to filter out outlier pleiotropic SNPs using "RadialMR" v1.0 R package[65]. The Cochran's Q test and Rucker's Q' test were performed to assess horizontal and directional pleiotropy effects with parameters "ivw_radial(alpha = 0.05, weight = 1, tol = 0.0001)" and "egger_radial(alpha = 0.05, weights = 1)", respectively. The outliers identified with a $P$ threshold of 0.05 were discarded. In addition, we calculated $F$-statistic[66] to measure the power of IVs with the formula $F = R^2(N - k - 1)/k(1 - R^2)$, where $N$ represents the sample size of exposure; $k$ represents the number of IVs; $R^2$ represents the proportion of phenotypic variance explained by the genetic variants, and is the sum of explained phenotypic variance of each variant. $R^2$ was calculated from effect size ($\beta$) and standard error (SE) using the formula $R^2 = \beta^2/(\beta^2 + SE^2 \times N)$. To avoid instrument bias, the $F$-statistic should be at least 10[66].

**Two-sample MR.** Associations of the genetic instruments (obesity associated variants in this study) identified in each brain cell type with diseases were estimated by performing two-sample MR analyses. Complementary methods based on different assumptions were used to find robust associations, including the inverse-variance weighting (IVW)[67], MR-Egger[68], weighted median[69], MR robust adjusted profile score (MR-RAPS)[70] and weighted mode[71]. The IVW method assumes balanced pleiotropy and implements multiplicative random-effect model to provide consistent estimates on causal effects[67]. The MR-Egger method estimates the causal effect through the slope coefficient of the Egger regression, which based on the Instrument Strength Independent of Direct Effect (InSIDE) assumption[68]. The MR-RAPS method leverages a robust adjusted profile score to accomplish statistical inference, which is also biased by unbalanced pleiotropy[70]. The weighted median method estimates the causal effect under the

assumption that at least 50% of the total weight of the instrument comes from valid variants[69]. The weighted mode provides consistent estimates assuming that the largest group of SNPs are valid instruments[71]. The Wald ratio method[72] is used to estimate causal effect when there is only one available genetic instrument. We used "TwoSampleMR" v0.5.6 R package to conduct these MR analyses. In this study, a robust causal inference was identified after Bonferroni correction for 180 MR tests (18 disease outcomes × (8 brain cell types + 2 brain tissues); $P < 2.78 \times 10^{-4}$ (0.05/180)).

**Pleiotropy and sensitivity analyses.** Among the results of six aforementioned MR analyses, we further performed a series of pleiotropy analyses to determine the main MR method. Firstly, we applied MR-PRESSO to detect the presence of horizontal pleiotropy (the MR-PRESSO global test)[73]. Then, we conducted MR-Egger regression to assess the potential directional pleiotropy according to the intercept term[68]. We also provided the calculation of Cochran's $Q$ and Rucker's $Q'$ statistics that were used to evaluate heterogeneity in the fixed-effect variance weighted analysis and MR-Egger analysis, respectively[74,75]. As the flow chart shown in Supplementary Fig. 6, We chose the main MR method as follows:

(a) If there was single genetic instrument, Wald ratio was used to estimate causal effect.

(b) For multiple genetic instruments, if no directional pleiotropy was detected, i.e., $P > 0.05$ for tests of Cochran's Q, MR-Egger intercept and MR-PRESSO, we chose IVW as the main method considering it is more powerful in detecting causal effects than other methods[71].

(c) In the case when directional pleiotropy was detected, the assumption of IVW and MR-RAPS was violated, whereas the MR-Egger was considered as the main MR method if $P > 0.05$ for the test of Rucker's Q'.

(d) If directional pleiotropy was detected and $P < 0.05$ for the test of Rucker's Q', the weighted median and weighted mode can be used to estimate causal effects, though the weighted median is more preferred given its higher power in detecting a causal effect[71].

We finally estimated the robustness of MR results by first performing leave-one-out sensitivity analysis to assess whether a single variant was driving the causal association. All the pleiotropy and sensitivity results are summarized and output in a separate file by csMR, which also includes a recommendation to help users select proper MR results.

**Reporting summary**

Further information on research design is available in the Nature Portfolio Reporting Summary linked to this article.

## Data availability

The summary statistics data for BMI and WHRadjBMI can be accessed from the GIANT data portal(https://portals.broadinstitute.org/collaboration/giant/index.php/GIANT_consortium_data_files). The body fat percentage GWAS data can be obtained from the MRC IEU OpenGWAS database (https://gwas.mrcieu.ac.uk/datasets/ukb-b-8909/). Brain single-cell eQTL data were provided by Bryois et al[7]. and accessed from https://doi.org/10.5281/zenodo.5543734. GTEx v8 data were downloaded from https://www.gtexportal.org/home/datasets. All GWAS summary statistics for disease outcomes used in this study are publicly available for download from the resources summarized in Supplementary Data 4.

## Code availability

The code used to perform the analysis described in this study is available at https://github.com/rhhao/csMR (csMR-v1.2[76]).

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

## Acknowledgements

We would like to thank the UK Biobank and GTEx Consortium for the reference genome and eQTL data. We obtained UK Biobank data under application number 46387. This work was supported by grants from the National Natural Science Foundation of China (32100416 (R.H.H.), 32370653 (Y.G.), 32170616 (T.L.Y.), 32070588 (S.S.D.)), Science Fund for Distinguished Young Scholars of Shaanxi Province (2021JC-02 (T.L.Y.)), Innovation Capability Support Program of Shaanxi Province (2022TD-44 (T.L.Y.)), China Postdoctoral Science Foundation (2021M702618 (R.H.H.)), Key Research and Development Project of Shaanxi Province (2022GXLH-01-22 (T.L.Y.)), Fundamental Research Funds for the Central Universities, and Medical-engineering Interdisciplinary Project of Xi'an Jiaotong University (YGJC202202 (T.L.Y.)). This study was also supported by the High-Performance Computing Platform and Instrument Analysis Center of Xi'an Jiaotong University. Parts of the Figs. 1, 3–4 were drawn by using pictures from Servier Medical Art and Scidraw.io, and changes were made to the pictures. Both Servier Medical Art and Scidraw.io are licensed under a Creative Commons Attribution 4.0 License (https://creativecommons.org/licenses/by/4.0/).

## Author contributions

R.H.H., S.S.D. and T.L.Y. conceived this study. T.L.Y and Y.G. organized and supervised the overall project. R.H.H., T.P.Z. and F.J. conducted the computational work. T.P.Z. and M.L. contributed to summary data collection. S.S.D. participated in data analysis. R.H.H. and J.H.L. prepared the tables and figures. R.H.H. and Y.G. wrote the manuscript with the assistance of other authors.

## Competing interests

The authors declare no competing interests.
