## [Peer Review File · Nature Communications]

Revealing brain cell-stratified causality through dissecting causal variants according to their cell-type-specific effects on gene expressionREVIEWER COMMENTS

Reviewer #1 (Remarks to the Author):

The noteworthy results of the study include the identification of cell-type-specific effects of genetic variants. They proposed an analytical framework called Cell-Stratified MR (csMR) to examine colocalized GWAS signals with single-cell expression quantitative trait loci (eQTL) from different brain cells. The developed framework csMR might be significant to the field of genetic causal association, which has the potential to reveal cell-type-specific causal effects.

However, some aspects of the paper remain unclear:

1. The precise number of genetic instrumental variables that successfully passed through the quality control measures remains undisclosed.
2. On line 82, the term "PPH4" should ideally be defined prior to its first usage to enhance clarity and understanding.
3. It remains uncertain what substantiates the assertion made on line 376, where a $PPH4 > 0.8$ is stated to be indicative of a colocalization signal. Further clarification or supporting information may be necessary to justify this claim.

Reviewer #2 (Remarks to the Author):

Yang and his colleagues developed an analytical framework, csMR, to find cell-stratified causality by integrating summary-level data of GWAS and single-cell eQTL. After applying it to BMI GWAS data, they demonstrated the value of applying csMR to help prioritize target cells to infer causality between BMI and 18 disease outcomes, which is often hidden from bulk analyses. csMR methodology is an innovative framework for revealing cellular hierarchical causality and will contribute to a better understanding of cell-specific genetic links between complex diseases. I have the following comments on this manuscript.

- 1) In two-sample MR analysis, six complementary methods were chosen to find robust associations. The different complementary methods are based on different assumptions, and there often some discrepancies in their association results. So, how to ensure that these assumptions are not violated in different data scenarios? How can these results be properly reconciled? Was it sufficient to simply merge

or intersect the results of different methods? It might be worthwhile to use some ensemble learning algorithm to merge these correlation results.

2) Brain single-cell eQTL data from previous studies, including cis- and trans-eQTL, were fed to the two-sample MR analysis. It would be interesting to evaluate the identified cell type-stratified variants in cis- and trans-eQTL groups separately. Do they have cell-type context-dependent regulatory effects on genes they regulate?

3) Although other obesity-related phenotypes such as WHR and fat mass are highly correlated BMI, they exhibited different cell-stratified causality to BMI. It seems a bit unexpected, how to interpret the biological significance of these results?

Response to reviewers

Reviewer #1 (Remarks to the Author):

The noteworthy results of the study include the identification of cell-type-specific effects of genetic variants. They proposed an analytical framework called Cell-Stratified MR (csMR) to examine colocalized GWAS signals with single-cell expression quantitative trait loci (eQTL) from different brain cells. The developed framework csMR might be significant to the field of genetic causal association, which has the potential to reveal cell-type-specific causal effects.

Response: We thank the reviewer for the effort reviewing our manuscript, and we also appreciate the reviewer's recognition of the significance of our work.

However, some aspects of the paper remain unclear:

Comments: 1. The precise number of genetic instrumental variables that successfully passed through the quality control measures remains undisclosed.

Response: Thanks for your comment. The number of genetic instrumental variables (IVs) that passed quality control tests was summarized in Supplementary Tables 4-5,7,9,11 at the first submission. In the current version, we have added this information to the main text. Please see the marked-up part on pages 7-10. Thank you!

Comments: 2. On line 82, the term "PPH4" should ideally be defined prior to its first usage to enhance clarity and understanding.

Response: We apologize for the lack of clarity. We have added detailed definition of "PPH4" in the current manuscript and made a full explanation in the Method section. Please see lines 81-82 and 390-394. Thank you!

Comments: 3. It remains uncertain what substantiates the assertion made on line 376, where a $PPH4 > 0.8$ is stated to be indicative of a colocalization signal. Further clarification or supporting information may be necessary to justify this claim.

Response: Thank you for your comment. " $PPH4 > 0.8$ " is a commonly used threshold for the evidence of strong colocalization with shared causal variants. As the reviewer suggested, we have cited supporting references in the current manuscript. Please see page 17, lines 394-395. Thank you!

Reviewer #2 (Remarks to the Author):

Yang and his colleges developed an analytical framework, csMR, to find cell-stratified causality by integrating summary-level data of GWAS and single-cell eQTL. After applying it to BMI GWAS data, they demonstrated the value of applying csMR to help prioritize target cells to infer causality between BMI and 18 disease outcomes, which is often hidden from bulk analyses. csMR methodology is an innovative framework for revealing cellular hierarchical causality and will contribute to a better understanding of cell-specific genetic links between complex diseases. I have the following comments on this manuscript.

Response: We thank the reviewer for recognizing the value of our work. We also appreciate your constructive comments, which have greatly improved the manuscript. Based on your suggestions, we have made careful and substantial revisions. We hope our response would address all your concerns.

Comments: 1) In two-sample MR analysis, six complementary methods were chosen to find robust associations. The different complementary methods are based on different assumptions, and there often some discrepancies in their association results. So, how to ensure that these assumptions are not violated in different data scenarios? How can these results be properly reconciled? Was it sufficient to simply merge or intersect the results of different methods? It might be worthwhile to use some ensemble learning algorithm to merge these correlation results.

Response: Thanks for your comment. As the reviewer mentioned, the six MR methods are based on different assumptions about horizontal pleiotropy and evaluate different statistical estimates, which is difficult to be merged. Whereas the violation of these assumptions can be recognized by performing additional pleiotropy analyses. In the revised manuscript, we attempted to select the proper MR method according to a series of pleiotropy analyses (as shown below).

We chose the main MR method as follows:

- (a) If there was single genetic instrument, Wald ratio was used to estimate causal effect.
- (b) For multiple genetic instruments, if no directional pleiotropy was detected, i.e., $P > 0.05$ for tests of Cochran's Q, MR-Egger intercept and MR-PRESSO, we chose IVW as the main method considering it is more powerful in detecting causal effects than other methods [1].
- (c) In the case when directional pleiotropy was detected, the assumption of IVW and MR-RAPS was violated, whereas the MR-Egger was considered as the main MR method if $P > 0.05$ for the test of Rucker's Q'.
- (d) If directional pleiotropy was detected and $P < 0.05$ for the test of Rucker's Q', the weighted median and weighted mode can be used to estimate causal effects, though the weighted median is more preferred given its higher power in detecting a causal effect [1].

We also implemented leave-one-out sensitivity analysis before reporting significant results. We have revised the manuscript to explain how we choose the main MR method in detail (see pages 19-20, lines 434-443 and 447-468), and added the flow chart to the supplementary file (see Supplementary Fig. 6). The main MR method we suggested has been included in Supplementary Tables 6,8,10,12. We also modified csMR to include the MR selection process and provide a recommendation in the output file. Thanks for your constructive comment!

1. Hartwig, F. P. *et al.* Robust inference in summary data Mendelian randomization via the zero modal pleiotropy assumption. *Int. J. Epidemiol.* **46**, 1985–1998 (2017).

Comments: 2) Brain single-cell eQTL data from previous studies, including cis- and trans-eQTL, were fed to the two-sample MR analysis. It would be interesting to evaluate the identified cell type-stratified variants in cis- and trans-eQTL groups separately. Do they have cell-type context-dependent regulatory effects on genes they regulate?

Response: Thank you for your valuable suggestion. We agree with the reviewer that it would be quite interesting to detect cell-stratified causality using both cis- and trans-eQTL separately. Unfortunately, the brain single-cell eQTL data only included cis-eQTL, while the trans-eQTL remains unreleased. It is worth mentioning that our csMR workflow also accepts trans-eQTL data with no need to change any scripts. We are very willing to perform further exploration once the single-cell trans-eQTL data is released. We have discussed this limitation in the current manuscript. Please see page 14, lines 313-318. Again, thanks for your comment!

Comments: 3) Although other obesity-related phenotypes such as WHR and fat mass are highly correlated BMI, they exhibited different cell-stratified causality to BMI. It seems a bit unexpected, how to interpret the biological significance of these results?

Response: Thanks for your comment. Our results showed different cell-stratified causality between obesity related traits, which reflects distinct cell effects and different causal pathways. This observation can be explained by the fact that these traits reflect different aspects of body composition, which was proven to be affected by distinct molecular processes and metabolic

mechanisms. BMI reflects overall adiposity but does not distinguish fat mass from lean mass; WHR cares more about abdominal and gluteofemoral fat; body fat percentage directly measures the amount of fat. Previous GWAS studies showed that the majority of associated loci and affected genes were unique between these traits [2-4]. Enrichment analyses with GWAS loci also revealed distinct metabolic pathways, which BMI-associated loci were significantly enriched for appetite regulation, whereas pathways involved in alterations in fat cell size and body fat amount were identified by body fat percentage associated loci [5]. In addition, studies also showed that the associated proteomic profiles were unique between BMI and WHR [6-7], indicating distinguishing molecular processes. Therefore, the distinct biological processes activated in specific cell might contribute to the cell-stratified causal effects on different disease outcome.

In order to make this point clearer, we have revised the corresponding paragraph in the current manuscript. Please see pages 13-14, lines 299-300, 304-312. Thank you!

2. Locke, A.E. *et al.* Genetic studies of body mass index yield new insights for obesity biology. *Nature* **518**, 197-206 (2015).
3. Shungin, D. *et al.* New genetic loci link adipose and insulin biology to body fat distribution. *Nature* **518**, 187-196 (2015).
4. Winkler, T.W. *et al.* A joint view on genetic variants for adiposity differentiates subtypes with distinct metabolic implications. *Nat Commun* **9**, 1946 (2018).
5. Ghosh, S. & Bouchard, C. Convergence between biological, behavioural and genetic determinants of obesity. *Nat Rev Genet* **18**, 731-748 (2017).
6. Lind, L. *et al.* Changes in Proteomic Profiles are Related to Changes in BMI and Fat Distribution During 10 Years of Aging. *Obesity (Silver Spring)* **28**, 178-186 (2020).
7. Bao, X. *et al.* Proteomic Profiles of Body Mass Index and Waist-to-Hip Ratio and Their Role in Incidence of Diabetes. *J Clin Endocrinol Metab* **107**, e2982-e2990 (2022).

REVIEWERS' COMMENTS

Reviewer #2 (Remarks to the Author):

The authors did an excellent job revising the manuscript. I have no further comments.